# Prevalence and Mutation Analysis of Medium-Chain Acyl-CoA Dehydrogenase Deficiency Detected by Newborn Screening in Hefei, China

**DOI:** 10.3390/ijns11030083

**Published:** 2025-09-22

**Authors:** Haili Hu, Qingqing Ma, Yong Huang, Wangsheng Song, Hongyu Xu, Peng Zhu, Yan Wang

**Affiliations:** 1Anhui Women and Children’s Medical Center, Hefei 230001, China; hhl919@aliyun.com (H.H.); mqq1099@163.com (Q.M.); topgun_hy@aliyun.com (Y.H.); sws8104@163.com (W.S.); yu56579849@126.com (H.X.); 2Department of Maternal, Child and Adolescent Health, School of Public Health, Anhui Medical University, Hefei 230022, China; pengzhu@ahmu.edu.cn

**Keywords:** medium-chain acyl-CoA dehydrogenase deficiency, octanoylcarnitine, ACADM gene, newborn screening

## Abstract

Medium-Chain Acyl-CoA Dehydrogenase Deficiency (MCADD) is a metabolic disorder caused by mutations in the ACADM gene, leading to impaired fatty acid oxidation. The present study aims to analyze the prevalence and genetic mutation characteristics of MCADD among newborns in Hefei, China, providing insights for the diagnosis, treatment, and prevention of MCADD. A retrospective analysis was conducted on data from newborns diagnosed with MCADD at the Hefei Newborn Disease Screening Center between January 2016 and December 2024. Screening was performed using tandem mass spectrometry (MS/MS), complemented by next-generation sequencing (NGS) for genetic testing. Out of 880,224 screened newborns, 16 cases of MCADD were diagnosed, resulting in a prevalence of 1 in 55,014. A total of 31 mutation sites in the *ACADM* gene were identified, with 18 different mutation types. The hotspot mutations were c.449-452del (p.T150Rfs*4) and c.1085G>A (p.G362E), each with a mutation frequency of 16.13% (5 out of 31). Additionally, three novel mutations were identified: c.468+5G>A, c.854C>G, and c.428_431delinsTCTTCTTTTGTT. Following diagnosis, patients received health education, dietary guidance, and symptomatic treatment, all resulting in favorable prognoses without any acute metabolic decompensation events. The prevalence of MCADD is lower in Asia compared to Europe and America. The hotspot mutations for MCADD in Hefei are c.449-452del and c.1085G>A. Diagnosis should integrate results from both octanoylcarnitine (C8) levels and genetic testing. Early screening, diagnosis, treatment, and scientific prevention strategies are essential for reducing adverse outcomes in children with MCADD.

## 1. Introduction

Medium-chain acyl-CoA dehydrogenase deficiency (MCADD) is a disorder caused by a functional defect in medium-chain acyl-CoA dehydrogenase, leading to impaired β-oxidation of medium-chain fatty acids. This results in decreased energy production and accumulation of toxic metabolic intermediates. MCADD has an insidious onset and is associated with a high mortality rate; approximately 25% of affected children succumb to the condition, while mortality in adults experiencing acute episodes can reach up to 50%. Patients often appear clinically normal during asymptomatic intervals. Among survivors of acute episodes, about one-third develop sequelae, including growth retardation, delayed motor development, intellectual disability, epilepsy, and cerebral palsy [1]. With the advancement and implementation of tandem mass spectrometry (MS/MS), many countries and regions have incorporated MCADD into newborn screening programs. Screening is effective in patients with MCAD deficiency since early diagnosis reduces deaths and severe adverse events in children up to the age of 4 years [2]. A significant reduction in the incidence of acute encephalopathy episodes and a lower prevalence of intellectual disability were observed following the implementation of expanded neonatal screening [3].

The relationship between MCADD genotype and clinical phenotype remains unclear. Due to the complexity of contributing factors, patients with MCADD within the same family often exhibit varying clinical phenotypes. Individuals with mild biochemical phenotypes may nonetheless experience life-threatening symptoms [4]. Studies have shown that compared with other mutations (0–63%), patients with the c.985A>G (p.K304E) homozygous mutation have significantly reduced residual MCAD activity (0–8%) and poor prognosis, and some patients only manifest symptoms in adulthood [5]. Gene-gene and gene-environment interactions may influence the disease’s natural course, making genotype alone insufficient to accurately predict clinical phenotype or disease severity.

The present study analyzed the prevalence and genetic mutation characteristics of MCADD based on newborn screening data from Hefei City, China, from January 2016 to December 2024, providing insights for the diagnosis, treatment, and prevention of MCADD. Due to mutations located in intronic splice sites, regulatory regions, or involving deletions, as well as epigenetic influences, diagnosis requires comprehensive evaluation combining octanoylcarnitine (C8) levels and genetic testing results. Compared to healthy peers, individuals who survive the initial onset of MCADD utilize significantly more medical resources during the early years of life and frequently face challenges related to the condition throughout their lives [6]. Early screening, diagnosis, treatment, and scientific prevention strategies are critical to reducing adverse outcomes in patients with MCADD.

## 2. Materials and Methods

### 2.1. Study Population

Our study analyzed 880,224 newborns screened for diseases using tandem mass spectrometry in Hefei City, China, from January 2016 to December 2024. Sample collection followed the principle of informed consent, with all parents of affected children providing signed consent forms. Our study was approved by the Ethics Committee of Anhui Women and Children’s Medical Center (Approval No: 2024-011), in accordance with the ethical standards of the Helsinki declaration and its later amendments or comparable ethical standards.

### 2.2. Newborn Screening for MCADD

Three drops of neonatal heel blood were collected onto specialized blood collection filter paper 3 to 7 days after birth, following adequate breastfeeding, and allowed to dry naturally at room temperature before sending it for testing. The dried samples were then sent for analysis. Detection was performed using a Waters TQD tandem mass spectrometer (Waters Corporation, Milford, MA, USA) employing liquid chromatography-tandem mass spectrometry (LC-MS/MS). The assay utilized non-derivatized kits for the quantification of multiple amino acids, carnitines, and succinylacetone, manufactured by PerkinElmer Company, Turku, Finland. Based on screening data from more than 300,000 newborns across multiple domestic regions using reagents from the same PerkinElmer company, as well as data collected from 20,000 newborns in the Hefei region between July and December 2015, the 0.5th and 99.5th percentiles were, respectively, utilized as the lower and upper limits of the analyte reference range. Accordingly, the reference interval in the present study was established as follows: free carnitine (C0), 10–55 μmol/L; C8, 0.02–0.15 μmol/L; and the C8/C10 ratio, 0.44–1.33. Elevated C8 levels or a concurrent increase in the C8/C10 ratio were considered suspicious, prompting recall for repeat tandem mass spectrometry testing. Persistent elevation upon retesting led to subsequent genetic testing.

### 2.3. Genetic Analysis

Peripheral venous blood (2 mL) was collected from neonates and their parents into EDTA anticoagulant tubes. Genomic DNA was extracted using a DNA isolation kit. Following the construction of a preliminary library with a commercial kit, targeted exon capture technology was employed to obtain the final sequencing library. Subsequently, next-generation high-throughput sequencing (NGS) was performed, encompassing the exons and adjacent intronic regions (20 base pairs upstream and downstream) of 85 genes, including *ACADVL*, *ACADM*, *ACADS*, *ETFA*, *ETFB*, and *ETFDH*. Suspected variant loci identified in the proband and their parents were validated through Sanger sequencing. The pathogenicity of the identified variants was assessed according to the 2015 guidelines of the American College of Medical Genetics and Genomics (ACMG), classifying variants into five categories: pathogenic, likely pathogenic, variants of uncertain significance (VUS), likely benign, and benign. Variants classified as pathogenic, likely pathogenic, or VUS were included in this study.

### 2.4. Screening and Diagnostic Criteria

Suspicious cases are defined by elevated levels of C8 or combined C8/C10, whereas confirmed cases are identified by the detection of ACADM gene mutations or by persistent elevation of C8 or combined C8/C10.

### 2.5. Statistical Analysis

Statistical analyses were performed using SPSS version 26.0 (SPSS Inc., Chicago, IL, USA). Data that followed a normal distribution are presented as mean ± standard deviation. The t-test was utilized to compare differences between the MCADD and non-MCADD groups, with a significance level set at *p* < 0.05, indicating statistical significance (α = 0.05).

## 3. Results

### 3.1. Baseline Characteristics

A total of 880,224 newborns were screened, yielding 254 suspected positive cases, corresponding to a suspected positive rate of approximately 0.029%. Among these, three newborns refused re-examination, and one underwent re-examination at an external hospital, which yielded normal results but without submission of the report. The screening center recalled 250 newborns for re-examination, of whom 26 tested positive again. Among these, one newborn, treated with high medium-chain triglyceride (MCT) formula due to chylothorax, had a mother who declined further examination. This newborn’s initial screening levels of C8 and C10 were 0.36 μmol/L and 0.06 μmol/L, respectively; re-screening levels were 0.33 μmol/L and 0.06 μmol/L, respectively. Additionally, two other newborns refused further examination: one had initial C8 and C10 levels of 0.34 μmol/L and 0.51 μmol/L, and re-screening levels of 0.46 μmol/L and 0.51 μmol/L; the other had initial levels of 4.82 μmol/L and 0.25 μmol/L, and re-screening levels of 1.45 μmol/L and 0.1 μmol/L. Based on biochemical markers, this newborn could be clinically diagnosed with MCADD but was lost to follow-up and thus excluded from our study. Fifteen newborns with elevated values underwent immediate genetic testing, all of whom were diagnosed with MCADD. Eight newborns were retested using tandem mass spectrometry after one month: seven exhibited normalization of C8 and other indicators, while one newborn (case 12) still showed elevated C8 and other indicators, underwent genetic testing, and was diagnosed with MCADD. During follow-up, fluctuations in C8 levels were observed; except for cases 4 and 9, who each had one normal measurement, all others consistently exhibited elevated levels above the normal range (Table 1).

A total of 16 MCADD cases were identified, including 6 males and 10 females, resulting in a prevalence rate of 1 in 55,014. At the initial screening of 16 cases, the mean concentration of C0 was 29.13 ± 4.94 μmol/L, all within the normal range. The mean concentration of C6 was 0.63 ± 0.42 μmol/L, exceeding the reference range in all cases. C8 averaged 3.30 ± 2.73 μmol/L, more than twice the upper limit of the reference range. The mean C10 concentration was 0.31 ± 0.21 μmol/L, with 9 cases within the normal range. C10:1 averaged 0.41 ± 0.27 μmol/L, with 2 cases within the normal range. The C8/C10 ratio was 11.19 ± 5.63, indicating significant elevation. Upon re-screening, C0 decreased significantly to 23.97 ± 5.93 μmol/L, remaining within the normal range for all cases (*p* < 0.05). C6 also decreased significantly to 0.40 ± 0.14 μmol/L but remained above the reference range (*p* < 0.05). C8 concentration was 1.43 ± 0.77 μmol/L, still elevated but significantly lower than at initial screening (*p* < 0.05). The mean C10 concentration decreased significantly to 0.13 ± 0.04 μmol/L, all values falling within the normal range (*p* < 0.05). C10:1 measured 0.27 ± 0.11 μmol/L, with 4 cases within the normal range, showing no statistically significant difference from the initial screening (*p* > 0.05). The C8/C10 ratio remained significantly elevated at 11.23 ± 5.23, with no significant change compared to initial screening (*p* > 0.05) (Table 2, Figure 1 and Figure 2).

### 3.2. Genetic Analysis of Newborns with MCADD

The study involved genetic testing of 16 pediatric patients, of whom 14 exhibited compound heterozygous mutations, 1 presented with a homozygous mutation (c.449-452del [p.T150Rfs4]), and 1 patient was identified with a single mutation site. During subsequent follow-up, consistently elevated levels of C8 and C8/C10 were observed, and high-throughput sequencing did not reveal any additional genetic metabolic disorders associated with elevated C8 (including variations in the *ETFA*, *ETFB*, and *ETFDH* genes). A total of 31 mutation sites were identified, encompassing 18 different mutation types, including 9 missense mutations, 3 deletion mutations, 3 frameshift mutations, 2 splice site mutations, and 1 synonymous mutation. These mutations were distributed across exons 1, 2, 5, 6, and 8–12, as well as in introns 4 and 6. The hotspot mutations identified were c.449-452del (p.T150Rfs4) and c.1085G>A (p.G362E), both exhibiting a mutation frequency of 16.13% (5/31). Three novel mutations were discovered and interpreted according to the ACMG guidelines: (1) c.468+5G>A was classified as a variant of uncertain significance (PM2+PP3), (2) c.854C>G (p.A285G) was classified as a variant of uncertain significance (PM2), and (3) c.428_431delinsTCTTCTTTTGTT (p.K143Ifs*10) was classified as a pathogenic variant (PVS1+PM2_P+PP4) (Table 3, Figure 3 and Figure 4).

### 3.3. Clinical Follow-Up Outcomes

All diagnosed pediatric patients received dietary guidance and health education, emphasizing the importance of avoiding fasting and adopting a low-fat diet. Case 1 exhibited decreased free C0 levels at 5 months of age and was prescribed oral L-carnitine at a dosage of 50–100 mg/kg/day. After one month of treatment, C0 levels normalized, and oral L-carnitine was continued. At 28 months, C0 levels declined again; after increasing the dosage, C0 returned to normal. Case 2 demonstrated decreased C0 levels, mildly elevated creatine kinase (CK), and CK isoenzyme levels at 5 months. Oral L-carnitine treatment similarly restored C0 and CK levels to normal, although CK isoenzymes remained mildly elevated. This patient exhibited no abnormal clinical symptoms. At 34 months, C0 decreased again, and after dosage escalation, C0 normalized while CK isoenzymes remained mildly elevated. Case 5 presented with decreased C0 at 6 months and received oral L-carnitine; C0 levels normalized after one month. After discontinuing L-carnitine at 26 months, C0 decreased again but normalized upon resumption of therapy. This patient also experienced a single episode of mild hyperammonemia at 18 months and sinus arrhythmia at 30 months. Cases 3 and 8 showed C0 concentrations below 15 μmol/L at 5 and 17 months, respectively, and were prophylactically treated with L-carnitine upon parental request. Case 9 exhibited mild elevation of CK isoenzymes at 12 months and was found to have a small pericardial effusion on echocardiography; no specific treatment was administered, and subsequent follow-up indicated normalization. Case 3 was diagnosed with mild tricuspid regurgitation via echocardiography at four years of age and is currently under ongoing surveillance. All patients were advised to avoid fasting during illness, maintain adequate caloric intake, and undergo regular monitoring of blood glucose, creatine kinase, C0, and C8. To date, no biochemical abnormalities, such as hypoglycemia, have been observed in any patient.

## 4. Discussion

The prevalence of MCADD varies significantly among ethnic groups, with higher incidence rates reported in Caucasians. The incidence rate among newborns in Europe ranges from 1 in 27,000 to 1 in 10,000 [7]. In Denmark, the rate is approximately 1 in 8954 [8]; in Germany, 1 in 9773 [9]; in the United Kingdom, around 1 in 10,000 [10]; and in the United States, between 1 in 13,000 and 1 in 19,000. In the Nordic countries, MCADD is recognized as the second most common inherited metabolic disorder after hyperphenylalaninemia. Conversely, the prevalence among Asian populations is considerably lower; in Japan, it is about 1 in 129,000 based on NBS results of 3.36 million newborns [11]. A decade-long newborn screening using tandem mass spectrometry in Zhejiang Province, China, reported an MCADD prevalence of 1 in 222,902 [12], and similar screening in another Chinese province indicated a prevalence of 1 in 263,500 [13]. Screening in Qingdao, involving over 270,000 newborns, found a prevalence of 1 in 69,545 [14], while in Zibo, screening of more than 240,000 newborns revealed a prevalence of approximately 1 in 40,261 [15]. Our center, after nine years of tandem mass spectrometry newborn screening encompassing over 880,000 neonates, confirmed 16 cases, corresponding to an MCADD prevalence of approximately 1 in 55,014 in Hefei city. This rate is notably higher than those reported in the previously mentioned Chinese provinces and Zhejiang but is comparable to the prevalences observed in Qingdao, Zibo, and Japan.

The concentrations of C0, C6, C8, and C10 in MCADD patients upon rescreening were significantly lower than those observed during initial screening, whereas no significant differences were found in the C10:1 concentration and the C8/C10 ratio between the two screenings. Because the concentrations of C0, C10, and C10:1 may fall within normal ranges in some patients, C6, C8, and the C8/C10 ratio may serve as more reliable indicators for MCADD screening. Touw CM et al. reported a negative correlation between the C8/C10 ratio and residual MCAD enzyme activity. Notably, all newborns exhibiting symptoms in the neonatal period had a C8/C10 ratio ≥ 10 and residual MCAD enzyme activity below 1% in newborn screening tests [5]. Combining C8 concentration measurement with the C8/C10 ratio enhances the sensitivity and accuracy of MCADD screening [16].

MCADD is an autosomal recessive genetic disease attributed to mutations in the *ACADM* gene, which is located on chromosome 1p31.1 and comprises 12 exons. To date, over 160 mutation sites have been identified, with a predominance of missense mutations. Among the Northern European population, the most common mutation is c.985A>G (p.K304E), which is found in exon 11. Khalid et al. analyzed the ethnic backgrounds of patients homozygous for the c.985A>G mutation and discovered that approximately 94% were Caucasian, with no representation of Asian or African individuals, suggesting a mutation with a distinct ethnic origin [17]. In Arab populations, the most frequent mutation is c.362C>T [18], while in Japanese and Korean populations, the prevalent mutation is c.449-452del (p.T150Rfs4) [19,20]. In Zhejiang Province, China, the hotspot mutation is also c.449-452del, accounting for 25% of cases [4]. In Hefei city, the hotspot mutations include c.449-452del (p.T150Rfs4) and c.1085G>A (p.G362E), each occurring with a frequency of 16.13%, indicating that c.449-452del may be a common mutation among East Asian populations.

Certain mutations located within intronic splice regions, regulatory elements, or involving fragment deletions and epigenetic influences cannot be detected using conventional high-throughput sequencing methods. As a result, some patients with autosomal recessive inherited metabolic diseases may present with only one detectable mutation or none at all. For these patients, in addition to whole-genome sequencing and real-time quantitative PCR (q-PCR) for mutation identification, differential diagnosis may be informed by their unique biochemical phenotypes [4]. In the present study, patient 16 was found to carry only the c.449-452del (p.T150Rfs*4) mutation inherited from the father through high-throughput sequencing. During subsequent follow-up, persistent elevations of C8 and the C8/C10 ratio were noted. No additional genetic variants associated with metabolic disorders that cause increased C8 levels, such as mutations in the *ETFA*, *ETFB*, and *ETFDH* genes, were identified through high-throughput sequencing, leading to a final diagnosis of MCADD. Additionally, patient 3 exhibited a deletion in the 5′UTR and exon 10, while patient 10 displayed deletions spanning exons 1 through 10. Patients 14 and 15 presented with deletions of exons 11 and 12. Due to the size of these deletions, validation by Sanger sequencing was not feasible, leaving the origin of these deletions undetermined.

Patients with MCADD predominantly present clinical symptoms between 3 months and 3 years of age, although a minority manifest during the neonatal period or adulthood, and some remain asymptomatic [21]. The disease course may involve one or multiple episodes, typically precipitated by triggering factors, with prolonged fasting being the most common. Concurrent infectious diseases are also frequent precipitating factors. Nearly all mutations associated with MCADD exhibit thermal instability, which may underlie the increased susceptibility to acute metabolic crises during febrile infections. In early-onset cases, initial symptoms commonly include lethargy and vomiting, but may also present with seizures or respiratory distress, often rapidly progressing to coma or death. The mortality rate among affected children is relatively high, with approximately 25% succumbing during episodes; adult-onset acute presentations carry an even higher mortality rate, reaching up to 50%. Patients generally appear normal during inter-episode intervals. Among survivors of acute episodes, about one-third develop sequelae such as impaired growth and motor development, intellectual disability, language deficits, neurobehavioral disorders, epilepsy, and cerebral palsy.

Exclusively breastfed neonates with MCAD are at risk for early metabolic decompensation. As breastfeeding rates increase, close management of feeding difficulties is essential for all neonates awaiting NBS results [22]. Newborn screening for medium-chain acyl-CoA dehydrogenase deficiency has detected cases with a wide range of genotypes and biochemical abnormalities. Although most children do well, adverse outcomes have not been entirely avoided. Assessment of potential risk and determination of appropriate treatment remain a challenge [23]. Newborn screening for MCADD reduces morbidity and mortality at an incremental cost below the range for accepted health care interventions [24].

The genotype-phenotype correlation in MCADD is not well understood due to the complexity of influencing factors; thus, genotype alone is insufficient for accurately predicting clinical phenotype or disease severity. International literature indicates that approximately 60% of symptomatic patients carry the homozygous c.985A>G (p.K304E) mutation, which results in the substitution of lysine with glutamic acid at position 304 of the mature protein. This mutation causes complete protein misfolding and functional loss [1]. A foreign study described a pedigree with deletions in exons 11 and 12 involving four children: the first child died suddenly at 48 h of age from an unexplained cause at autopsy; the second pregnancy involved monozygotic twins, one of whom succumbed on day 2 due to hypoglycemia, while the other survived after resuscitation and was diagnosed with MCADD through decreased MCAD activity in cultured skin fibroblasts. The fourth child exhibited no hypoglycemia at 36 h but was diagnosed metabolically through fasting tests and confirmed by molecular detection of exon 11–12 deletions, including the c.985A>G mutation. The researchers therefore suggested that patients carrying the c.985A>G mutation have a poor prognosis [16]. Chinese researchers, including Dong Liping et al. [6], reported a case of a neonate with partial deletions in exons 11 and 12 who developed symptoms in the perinatal period. The mother was a primipara with feeding intervals exceeding 5 h. The infant exhibited hypoglycemia, hyperkalemia, hyperammonemia, and elevated hepatic and myocardial enzymes. Due to the insidious onset and ambiguous diagnosis during screening, resuscitation was unsuccessful, resulting in death. This case led the authors to infer that partial deletions of exons 11 and 12 may be a significant cause of early neonatal decompensation and death.

Review of the relevant literature, combined with data from this study, indicates that the c.449-452del (p.T150Rfs*4) mutation is relatively prevalent in East Asian populations, with approximately 60% of Japanese patients carrying this variant. Hara K. reported that MCADD patients harboring the c.449-452del mutation exhibit enzyme activity below detectable levels. A homozygous patient with this mutation experienced hypoglycemia and loss of consciousness at 13 months of age; notably, this patient was not identified through newborn screening. In contrast, the c.1085G>A (p.G362E) mutation, which has residual enzyme activity of approximately 1%, was detected only in newborns who screened positive for metabolic disorders, and no associated clinical symptoms have been observed [25].

In the present study, Cases 14 and 15 were both found to have exon 11–12 deletions; after follow-up to ages 6 and 4 years, respectively, no abnormal clinical manifestations were observed, potentially due to timely dietary guidance and health education post-diagnosis. Therefore, early screening, diagnosis, treatment, and scientific prevention are critical for reducing adverse outcomes in MCADD patients. In cases 1, 2, and 5, C8 levels decreased in parallel with reductions in C0 during follow-up, potentially indicating secondary carnitine deficiency; blood glucose levels remained within the normal range. In case 2, the decline in C0 was accompanied by simultaneous increases in creatine kinase and its isoenzymes. Following oral L-carnitine treatment, creatine kinase levels returned to normal, whereas creatine kinase isoenzymes remained mildly elevated; the patient exhibited no abnormal clinical symptoms. Case 3 presented with mild tricuspid valve regurgitation, while C0, C8, blood glucose, and creatine kinase levels were unremarkable. In case 9, an increase in creatine kinase isoenzymes was observed alongside a decrease in C8 compared to previous measurements, whereas C0 and blood glucose levels remained normal. Case 12 is a homozygote for the c.449-452del mutation and is currently 1 year and 3 months old, with normal growth and development. No episodes of acute metabolic imbalance were recorded throughout the study. The decrease in C0 may contribute to elevated creatine kinase, and symptomatic treatment with L-carnitine supplementation proved effective.

In conclusion, the prevalence of MCADD in Hefei, China, exceeds that observed in southern regions of the country. However, the prevalence of MCADD in Asia is lower compared to that in Europe and North America. The predominant mutations associated with MCADD in Hefei include c.449-452del and c.1085G>A. Furthermore, three novel mutations have been identified—c.468+5G>A, c.854C>G, and c.428_431delinsTCTTCTTTTGTT—thereby expanding the mutation spectrum of the *ACADM* gene. A comprehensive diagnosis should incorporate both C8 acylcarnitine levels and genetic testing results. Early screening, diagnosis, treatment, and preventative strategies are crucial for reducing the risk of adverse outcomes in children affected by MCADD.

## Figures and Tables

**Figure 1 IJNS-11-00083-f001:**
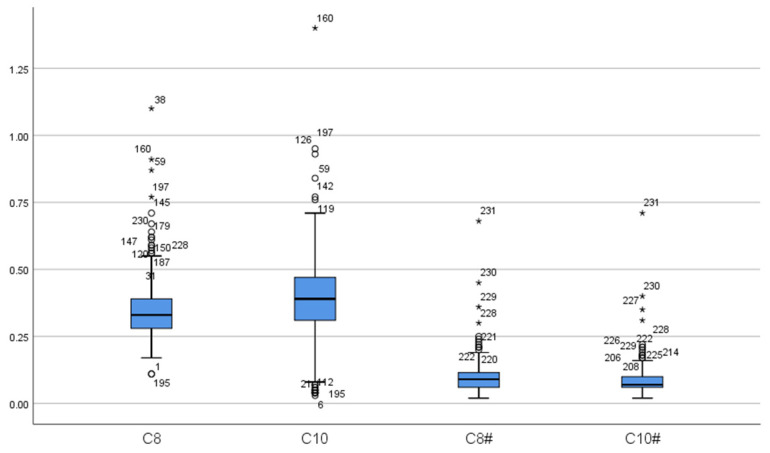
The distribution of C8 and C10. (C8: C8 in false-positive initial screening; C10: C10 in false-positive initial screening; C8#: C8 in false-positive repeat screening; C10#: C10 in false-positive repeat screening. “*” represent outliers, while “○” indicate extreme outliers).

**Figure 2 IJNS-11-00083-f002:**
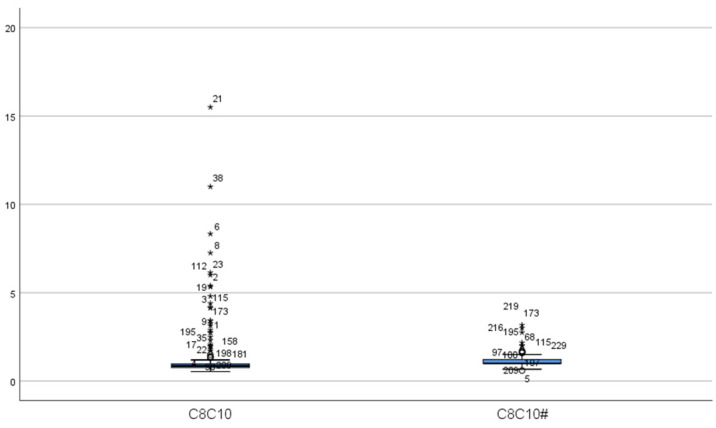
The distribution of C8/C10 ratio. (C8C10: C8/C10 in false-positive initial screening; C8C10#: C8/C10 in false-positive repeat screening. “*” represent outliers, while “○” indicate extreme outliers).

**Figure 3 IJNS-11-00083-f003:**
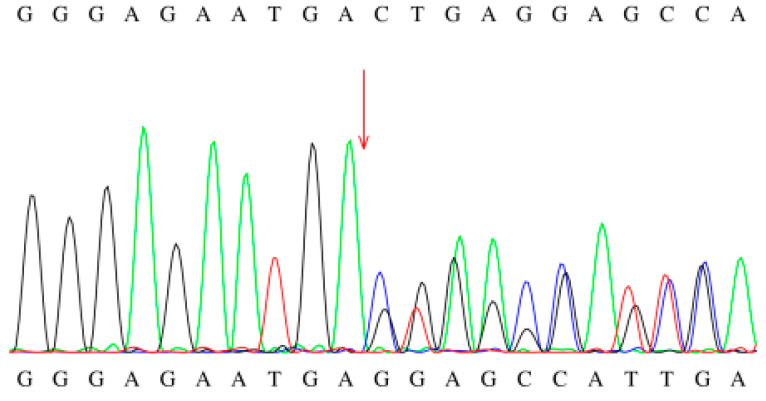
Sanger sequencing chromatogram of the c.449-452del mutation.

**Figure 4 IJNS-11-00083-f004:**
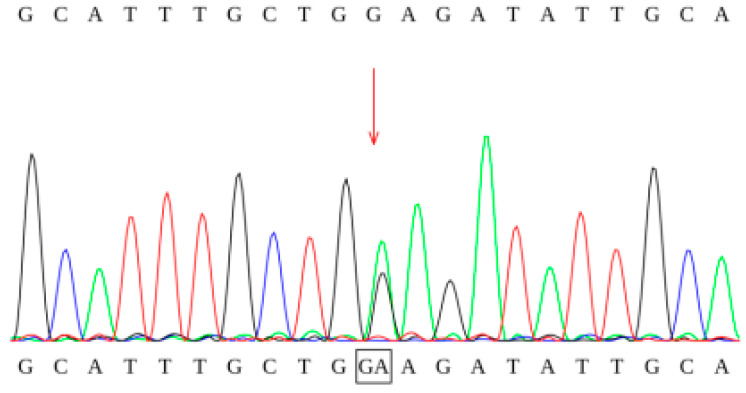
Sanger sequencing chromatogram of the c.1085G>A mutation.

**Table 1 IJNS-11-00083-t001:** The results of neonatal MCADD screening in Hefei from 2016 to 2024.

Case	Gender	Initial Screening			Secondary Screening			C8 Fluctuation Range (μmol/L) *	Other Abnormalities
		C0 (μmol/L)	C8 (μmol/L)	C8/C10	C0 (μmol/L)	C8 (μmol/L)	C8/C10		
1	Male	26.89	2.7	18	20.22	1.69	15.364	0.95–3.66	C0: 8.09
2	Male	33.17	11.3	17.12	24.97	2.12	15.143	0.95–11.3	C0: 6.57, CK: 219
3	Male	22.72	1.47	16.33	17.46	1.64	18.222	1.17–5.65	Cardiac ultrasound: mild tricuspid valve regurgitation
4	Male	35.01	0.64	3.05	30.81	0.64	3.765	0.09–1.05	
5	Female	35.59	5.75	10.27	19.46	0.73	10.429	0.22–5.75	C0: 9.94, blood ammonia: 77, sinus arrhythmia
6	Female	30.06	1.4	8.75	23.39	1.28	9.143	0.25–1.4	
7	Female	31.08	2.44	14.35	20.06	2.8	16.471	2.44–2.8	
8	Female	36.42	1.94	5.54	26.45	1.01	5.611	0.57–1.94	
9	Female	31.94	1.19	5.67	32.09	0.51	3.923	0.13–1.19	CK-MB: 48.1
10	Female	22.94	5.8	14.15	26.8	1.87	12.467	1.87–5.8	
11	Female	30.43	4.15	15.37	22.93	1.76	11.733	1.76–4.15	
12	Female	27.24	0.32	3.56	20.19	0.25	4.167	0.25–0.95	
13	Male	19.79	5.12	6.65	15.32	1.00	9.091	0.86–5.12	
14	Male	24.05	2.3	19.17	17.64	1.63	20.375	1.63–2.3	
15	Female	30.86	2.77	5.43	35.9	1.09	9.083	0.92–2.77	
16	Female	27.96	3.43	15.59	29.79	2.93	14.65	1.93–3.43	

* The “C8 fluctuation range” data were measured using DBS.

**Table 2 IJNS-11-00083-t002:** Comparison of initial and repeat screening acylcarnitine concentrations in children with MCADD.

	C0 (μmol/L)	C6 (μmol/L)	C8 (μmol/L)	C10 (μmol/L)	C10:1 (μmol/L)	C8/C10
Initial screening	29.13 ± 4.94	0.63 ± 0.42	3.30 ± 2.73	0.31 ± 0.21	0.41 ± 0.27	11.19 ± 5.63
Repeat screening	23.97 ± 5.93	0.40 ± 0.14	1.43 ± 0.77	0.13 ± 0.04	0.27 ± 0.11	11.23 ± 5.23
t	3.708	2.205	2.908	3.324	1.917	0.079
*p*	0.002	0.044	0.011	0.002	0.074	0.938

**Table 3 IJNS-11-00083-t003:** Genetic characteristics of newborns with MCADD.

Case	Mutation Site 1					Mutation Site 2				
	Location	Nucleotide Alteration	Amino Acid Alteration	Pathogenic Risk	Source	Location	Nucleotide Alteration	Amino Acid Alteration	Pathogenic Risk	Source
1	exon1	c.1A>G	p.M1V	Likely pathogenic	Mother	exon11	c.1085G>A	p.G362E	Pathogenic	Father
2	exon8	c.668T>C	p.I223T	Variant of uncertain significance	Mother	intron6	c.468+5G>A^#^	-	Variant of uncertain significance	Father
3	exon1	c.1A>G	p.M1V	Likely pathogenic	Mother	5′UTR-exon10	5′UTR-exon10del	-	Likely pathogenic	Not detected
4	exon2	c.91C>T	p.R31C	Variant of uncertain significance	Mother	exon10	c.854C>G^#^	p.A285G	Variant of uncertain significance	Father
5	exon11	c.1085G>A	p.G362E	Pathogenic	Mother	exon8	c.616C>T	p.R206C	Pathogenic	Father
6	exon11	c.1085G>A	p.G362E	Pathogenic	Mother	exon2	c.91C>T	p.R31C	Variant of uncertain significance	Father
7	intron4	c.286+232C>G	-	Variant of uncertain significance	Mother	exon5	c.383delG	p.L128Wfs*22	Pathogenic	Father
8	exon6	c.449-452del	p.T150Rfs*4	Pathogenic	Mother	exon11	c.1171A>G	p.M391V	Variant of uncertain significance	Father
9	exon6	c.428-431 delinsTCTTCTTTTGTT^#^	p.K143Ifs*10	Pathogenic	Mother	exon11	c.982A>G	p.M328V	Pathogenic	Father
10	exon1-exon10	exon1-exon10del	-	Pathogenic	Not detected	exon6	c.449-452del	p.T150Rfs*4	Pathogenic	Father
11	exon6	c.449-452del	p.T150Rfs*4	Pathogenic	Mother	exon6	c.449-452del	p.T150Rfs*4	Pathogenic	Father
12	exon2	c.33C>T	p.V11=	Variant of uncertain significance	Mother	exon6	c.428-431 delinsTCTTCTTTTGTT^#^	p.K143Ifs*10	Pathogenic	Father
13	exon11	c.1085G>A	p.G362E	Pathogenic	Mother	exon9	c.718A>G	p.M240V	Likely pathogenic	Father
14	exon11-exon12	exon11-exon12del	-	Likely pathogenic	Not detected	exon11	c.1085G>A	p.G362E	Pathogenic	Father
15	exon11	c.1171A>G	p.M391V	Variant of uncertain significance	Mother	exon11-exon12	exon11-exon12del	-	Likely pathogenic	Not detected
16	Not detected					exon6	c.449-452del	p.T150Rfs*4	Pathogenic	Father

"*" is for Protein translation termination and "^#^" is for New mutation.

## Data Availability

The datasets generated and analyzed during the current study are not publicly available due to participant privacy concerns but are available from the corresponding author on reasonable request.

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
