# Peer review of "Prevalence and Mutation Analysis of Medium-Chain Acyl-CoA Dehydrogenase Deficiency Detected by Newborn Screening in Hefei, China"

_2409-515X, 2025, doi:10.3390/ijns11030083_

Round 1
Reviewer 1 Report
Comments and Suggestions for Authors
The authors report data from a retrospective study of MCADD. Their work provides epidemiological insights into the city of Hefei. They also report new mutations and confirm the importance of newborn screening for a favorable prognosis for MCADD.
The paper is logically structured, and the work is quite interesting; however, some clarifications are needed.
1) The authors should explain how they calculated the normal range for the analytes reported and say which percentile of the population it represents. They should also confirm whether all newborns who underwent special procedures were excluded from the analysis.
2) Why didn't the authors report the C6, C10, and C10:1 values? These analytes are included in the kit they used, and they could be very useful in identifying true positives.
3) On line 100, the authors state that “diagnosis can be confirmed based on persistently elevated C8 levels”. However, it is important to note that medium-chain acylcarnitines have been observed to fluctuate. The same authors also report data showing a significant decrease in C8 concentration in secondary screening. Therefore, it would be prudent to exercise caution when checking only for C8, as there is a risk of overlooking a patient who exhibits a low C8 value in a secondary test or during follow-up. Why don't they perform the enzymatic test? Residual MCAD enzyme activity is crucial for establishing a diagnosis and for gaining a deeper understanding of the genotype-phenotype correlation.
4) How do the authors explain the strange condition of patient 16, which shows a high value of C8 and ratio but just one mutation? It looks like a heterozygote. In this case, it is very important to investigate the remaining enzymatic activity.
5) Incorporating further references would enhance the study, particularly concerning describing other observed prevalence rates in different geographical areas.
6) The names of genes should be written in Italics

Author Response
Dear Editors and Reviewers,
We are pleased to submit our revised manuscript entitled " Prevalence and mutation analysis of medium-chain acyl-CoA dehydrogenase deficiency detected by newborn screening in Hefei, China " (Manuscript Number: IJNS-3744416) for your consideration. We thank the editors and reviewers for their evaluation of this paper and their valuable comments. We have carefully considered and addressed each point, and have included a point-by-point response to the comments provided by reviewers. The comments are italicized and placed in square brackets. We have used the red text to highlight changes in response to the reviewers’ comments accordingly.
Reply to comments from Referee #1:
[Comments to the Author: The authors report data from a retrospective study of MCADD. Their work provides epidemiological insights into the city of Hefei. They also report new mutations and confirm the importance of newborn screening for a favorable prognosis for MCADD. The paper is logically structured, and the work is quite interesting; however, some clarifications are needed.]
Thank you for your time and efforts in reviewing our work. We appreciate your valuable comments and suggestions.
[1. The authors should explain how they calculated the normal range for the analytes reported
and say which percentile of the population it represents. They should also confirm whether all
newborns who underwent special procedures were excluded from the analysis.]
Thank you for your valuable suggestion. Given the skewed distribution of circulating C4 concentration and the regional differences in reference values for newborn screening, we employed the percentile method to establish the upper and lower limits for the cutoff values [Genet Med, 2011, 13(3):230-54]. The reference value ranges in our study were established based on screening data from more than 300,000 newborns across multiple domestic regions using reagents from the same PerkinElmer company, as well as data collected from 20,000 newborns in the Hefei region between July and December 2015. The lower and upper limits of the analyte reference range were defined as the 0.5th and 99.5th percentiles, respectively, of the normal population. The present study excluded other diseases that might cause elevated C8 levels, such as multiple acyl-CoA dehydrogenase deficiency. We have now clarified this point in the Materials and methods section (Materials and methods: line 90-94).
[2. Why didn't the authors report the C6, C10, and C10:1 values? These analytes are included
in the kit they used, and they could be very useful in identifying true positives. ]
We appreciate your valuable suggestion and comments. At the initial screening of 16 cases, the mean concentration of C0 was 29.13 ± 4.94 μmol/L, all within the normal range. The mean concentration of C6 was 0.63 ± 0.42 μmol/L, exceeding the reference range in all cases. C8 averaged 3.30 ± 2.73 μmol/L, more than twice the upper limit of the reference range. The mean C10 concentration was 0.31±0.21 μmol/L, with 9 cases within the normal range. C10:1 averaged 0.41 ± 0.27 μmol/L, with 2 cases within the normal range. The C8/C10 ratio was 11.19±5.63, indicating significant elevation. Upon re-screening, C0 decreased significantly to 23.97± 5.93μmol/L, remaining within the normal range for all cases (P < 0.05). C6 also decreased significantly to 0.40 ±0.14 μmol/L but remained above the reference range (P< 0.05). C8 concentration was 1.43 ± 0.77 μmol/L, still elevated but significantly lower than at initial screening (P < 0.05). The mean C10 concentration decreased significantly to 0.13 ± 0.04 μmol/L, all values falling within the normal range (P < 0.05). C10:1 measured 0.27 ± 0.11 μmol/L, with 4 cases within the normal range, showing no statistically significant difference from the initial screening (P > 0.05). The C8/C10 ratio remained significantly elevated at 11.23 ± 5.23, with no significant change compared to initial screening (P > 0.05). We have revised the Results section accordingly (Results: line 146-160).
[3. On line 100, the authors state that “diagnosis can be confirmed based on persistently elevated C8 levels”. However, it is important to note that medium-chain acylcarnitines have been observed to fluctuate. The same authors also report data showing a significant decrease in C8 concentration in secondary screening. Therefore, it would be prudent to exercise caution when checking only for C8, as there is a risk of overlooking a patient who exhibits a low C8 value in a secondary test or during follow-up. Why don't they perform the enzymatic test? Residual MCAD enzyme activity is crucial for establishing a diagnosis and for gaining a deeper understanding of the genotype-phenotype correlation.]
Thank you for the valuable suggestion. The patients' C8 concentrations during re-screening decreased significantly compared to the initial screening, while the C8/C10 ratio showed no difference between the two screenings. Therefore, detecting C8 concentration combined with the C8/C10 ratio can improve the sensitivity and accuracy of screening [Mol Genet Metab. 2014,113(4):274-7]. A study by Touw CM et al. found that the C8/C10 ratio is negatively correlated with residual MCAD enzyme activity; all newborns who exhibited symptoms during the neonatal period had a C8/C10 ratio ≥10 in newborn screening tests and residual MCAD enzyme activity below 1% [Orphanet J Rare Dis. 2012,7:30]. Since our study lacks the conditions to detect residual MCAD enzyme activity, we combine C8 concentration with the C8/C10 ratio for positive recall during screening. We have now clarified this point in the Materials and methods section (Materials and methods: line 115-117) and the Discussion sections (Discussion: line 227-237).
[4. How do the authors explain the strange condition of patient 16, which shows a high value of C8 and ratio but just one mutation? It looks like a heterozygote. In this case, it is very important to investigate the remaining enzymatic activity.]
Thank you for the valuable suggestion. In the present study, patient 16 was found to carry only the c.449-452del (p.T150Rfs*4) mutation inherited from the father through high-throughput sequencing. During subsequent follow-up, persistent elevations of C8 and the C8/C10 ratio were noted. No ad-ditional genetic variants associated with metabolic disorders that cause increased C8 levels, such as mutations in the ETFA, ETFB, and ETFDH genes, were identified through high-throughput sequencing, leading to a final diagnosis of MCADD. We have clarified this point in the Discussion sections (Discussion: line 258-264) .
[5. Incorporating further references would enhance the study, particularly concerning describing other observed prevalence rates in different geographical areas.]
Thank you for the valuable suggestion. We agree and have cited four additional references and compared their results with ours in the Discussion sections (Discussion: line 211-213).
[6. The names of genes should be written in Italics.]
We apologize for this error and have corrected it. We have revised the formatting of gene names to be italicized.

Reviewer 2 Report
Comments and Suggestions for Authors
This article describes Chinese patients with MCAD deficiency found by newborn screening (NBS) conducted in a region of mainland China. Consider revising the following points.
1. Introduction
The second paragraph (The present study retrospectively analyzed...) is not suitable for introduction. As this article mainly focused on genotypes, you should briefly summarize previous literature on phenotype-genotype information of NBS-detected cases compared with symptomatic cases.
2. Materials and methods
2.2. Newborn Screening for MCADD
It is essentially important to show when the first dried blood specimen are collected.
Do you mean that cutoff for C8 and C8/C10 were ≥0.15 and ≥1.33, respectively? Please add brief explanation about the statistical basis on which you set them.
3. Results
3.1. Basic Characteristics
Not only the data of confirmed cases but basic information on the results of NBS for MCADD must be revealed, such as total numbers of positive results and false-positive rate.
It is known that, in NBS for fatty acid disorders including MCADD, levels of marker acylcarnitines in generally decrease as feeding establishes, and retest of positive cases by dried blood specimen (DBS) is associated with the risk of false-negative results. Please show the distribution of C8 and C10 in the first and second DBS of the cases you concluded as false-positive.
Were the data of “C8 fluctuation range” in Table 1 measured in DBS or serum?
3.2. Genetic Anslysis of Newborns with MCADD, and 4. Discussion
In NBS for fatty acid disorders, functional analysis of variant enzymes is quite important for appropriate management, especially of those with novel variants. As this study did not have its own data on MCAD enzymatic assay, you should thoroughly refer to previous articles where such information are presented, especially to those on Asian populations. To show only ACMG classification in Table 2 is not satisfactory.
Among the data on disease frequency, the description “51,000 in Japan” is devoid of reference, and it is far higher than I know.
3.3. Clinical Follow-Up Outcomes
You should show the actual data of interest, namely C0 and CK, observed in the follow-up. Additional data of C8 and blood glucose on sick days will helpful for readers to estimate disease severity of your patients.
Author Response
Dear Editors and Reviewers,
We are pleased to submit our revised manuscript entitled " Prevalence and mutation analysis of medium-chain acyl-CoA dehydrogenase deficiency detected by newborn screening in Hefei, China " (Manuscript Number: IJNS-3744416) for your consideration. We thank the editors and reviewers for their evaluation of this paper and their valuable comments. We have carefully considered and addressed each point, and have included a point-by-point response to the comments provided by reviewers. The comments are italicized and placed in square brackets. We have used the red text to highlight changes in response to the reviewers’ comments accordingly.
Reply to comments from Referee #2:
[Comments to the Author: This article describes Chinese patients with MCAD deficiency found by newborn screening (NBS) conducted in a region of mainland China. Consider revising the following points.]
Thank you for your time and efforts in reviewing our work. We appreciate your valuable comments and suggestions.
[1. Introduction
The second paragraph (The present study retrospectively analyzed...) is not suitable for introduction. As this article mainly focused on genotypes, you should briefly summarize previous literature on phenotype-genotype information of NBS-detected cases compared with symptomatic cases.]
Thank you for pointing out this for us. We have now revised the Introduction accordingly (Introduction: line 53-65).
[2. 2. Materials and methods
2.2. Newborn Screening for MCADD
It is essentially important to show when the first dried blood specimen are collected.
Do you mean that cutoff for C8 and C8/C10 were ≥0.15 and ≥1.33, respectively? Please add brief explanation about the statistical basis on which you set them.]
We appreciate your valuable suggestion. In the present study, three drops of neonatal heel blood were collected onto specialized blood collec-tion filter paper 3 to 7 days after birth, following adequate breastfeeding. Given the skewed distribution of circulating C4 concentration and the regional differences in reference values for newborn screening, we employed the percentile method to establish the upper and lower limits for the cutoff values [Genet Med, 2011, 13(3):230-54]. The reference value ranges in our study were established based on screening data from more than 300,000 newborns across multiple domestic regions using reagents from the same PerkinElmer company, as well as data collected from 20,000 newborns in the Hefei region between July and December 2015. The lower and upper limits of the analyte reference range were defined as the 0.5th and 99.5th percentiles, respectively, of the normal population. We have clarified these points in the Materials and methods section (Materials and methods: line 84; line 90-94).
[3. Results
3.1. Basic Characteristics
Not only the data of confirmed cases but basic information on the results of NBS for MCADD must be revealed, such as total numbers of positive results and false-positive rate.
It is known that, in NBS for fatty acid disorders including MCADD, levels of marker acylcarnitines in generally decrease as feeding establishes, and retest of positive cases by dried blood specimen (DBS) is associated with the risk of false-negative results. Please show the distribution of C8 and C10 in the first and second DBS of the cases you concluded as false-positive.
Were the data of “C8 fluctuation range” in Table 1 measured in DBS or serum?]
Thank you for your valuable suggestions. From 2016 to 2024, a total of 880,224 newborns were screened, with 254 suspected positive cases, yielding a suspected positive rate of approximately 0.029%. Among them, 3 refused re-examination, and 1 was re-examined at an external hospital with normal results but did not provide the report. The new screening center recalled 250 for re-examination; 26 remained positive on re-screening. Among these, one newborn was treated with high MCT formula for chylothorax, and the mother refused further testing. This newborn’s initial screening C8 and C10 levels were 0.36 μmol/L and 0.06 μmol/L, respectively; on re-screening, C8 and C10 were 0.33 μmol/L and 0.06 μmol/L, respectively. Additionally, 2 other newborns refused further testing: one had initial screening C8 and C10 levels of 0.34 μmol/L and 0.51 μmol/L, and re-screening levels of 0.46 μmol/L and 0.51 μmol/L; the other had initial C8 and C10 at 4.82 μmol/L and 0.25 μmol/L, and re-screening at 1.45 μmol/L and 0.1 μmol/L. Based on biochemical indicators, this newborn could be clinically diagnosed with MCAD, but was lost to follow-up and not included in this study. Fifteen newborns with elevated values underwent immediate genetic testing and were all diagnosed with MCADD. Eight newborns were re-examined by tandem mass spectrometry after one month; seven showed normalized C8 and related indicators, while one (Case 12) still had elevated C8 and underwent genetic testing confirming MCADD. We have clarified these points in the Results section (Results: line 126-144).
Excluding 6 cases who refused re-examination and 1 case re-examined externally without a report, there were 231 cases determined false positive at this center, with a false positive rate of 0.026% (231/880,201). False positive cases initially showed C8 levels ranging from 0.11 to 1.1 μmol/L, C10 from 0.03 to 1.4 μmol/L, and C8/C10 ratios from 0.54 to 15.5 in dried blood spot testing. On re-screening dried blood spots, C8 ranged from 0.02 to 0.68 μmol/L, C10 from 0.02 to 0.71 μmol/L, and C8/C10 ratios from 0.6 to 3.17.
The "C8 fluctuation range" data presented in Table 1 were measured using DBS.
Note: C8: False positive primary screening C8; C10: False positive primary screening C10; C8#: False positive secondary screening C8; C10#: False positive secondary screening C10.
[3.2. Genetic Anslysis of Newborns with MCADD, and 4. Discussion
In NBS for fatty acid disorders, functional analysis of variant enzymes is quite important for appropriate management, especially of those with novel variants. As this study did not have its own data on MCAD enzymatic assay, you should thoroughly refer to previous articles where such information are presented, especially to those on Asian populations. To show only ACMG classification in Table 2 is not satisfactory.
Among the data on disease frequency, the description “51,000 in Japan” is devoid of reference, and it is far higher than I know.]
Thank you for the valuable suggestion. We agree that residual MCAD enzyme activity is important for gaining a deeper understanding of the genotype-phenotype correlation, however, our study lacks the facilities to perform residual MCAD enzyme activity testing. We reviewed previous articles on MCAD enzyme activity in Asian populations and integrated the related findings into the discussion section (Discussion: line 315-324).
We apologize for not including the reference for the cited "51,000 in Japan" earlier; we have now added the reference [Ref. 11] in the discussion section (Discussion: line 217).
[3.3. Clinical Follow-Up Outcomes
You should show the actual data of interest, namely C0 and CK, observed in the follow-up. Additional data of C8 and blood glucose on sick days will helpful for readers to estimate disease severity of your patients.]
Thank you for pointing out this for us. We have presented additional abnormalities, including C0 and CK, in Table 1. Furthermore, data on C8 and blood glucose levels for the cases have been included in the Results section (Results: line 203-208) and the Discussion section (Discussion: line 329-343).

Round 2
Reviewer 2 Report
Comments and Suggestions for Authors
This is a revised manuscript on newborn screening for MCAD deficiency conducted in Mainland China. I would like to ask the authors to revise several minor points, as follows.
3.1. Baseline Characteristics
Thank you for adding minute information (from line 127 to 161). Consider summarizing them in an additional table, which will make it easier to follow the description. The distribution of C8 and C8/C10 in the false-positive group (which you showed as a graph in your cover letter for this revision) should be included.
Table 1.
I understand that “the "C8 fluctuation range" data presented in Table 1 were measured using DBS”. You should show that in a footnote of the table.
- Discussion
The reference for the disease frequency in Japan as 1/51,000 (11. Shigematsu et al 2002) is too old. You should refer to the following article instead, which showed the frequency of 1/129,000 based on NBS results of 3.36 million newborns.
Shibata N, Hasegawa Y, et al. Diversity in the incidence and spectrum of organic acidemias, fatty acid oxidation disorders, and amino acid disorders in Asian countries: Selective screening vs. expanded newborn screening. Mol Genet Metab Rep 2018;16:5-10.
Author Response
Reply to comments from Reviewer 2:
[Comments to the Author: This is a revised manuscript on newborn screening for MCAD deficiency conducted in Mainland China. I would like to ask the authors to revise several minor points, as follows.]
Thank you for your time and efforts in reviewing our work. We have carefully considered and addressed each point, and have included a point-by-point response to the comments.
[3.1. Baseline Characteristics
Thank you for adding minute information (from line 127 to 161). Consider summarizing them in an additional table, which will make it easier to follow the description. The distribution of C8 and C8/C10 in the false-positive group (which you showed as a graph in your cover letter for this revision) should be included. ]
Thank you for your valuable suggestions. We agree and have now added an additional table (Table 2) in the Result section (Results: line 166). We have also included a graph to illustrate the distribution of C8 and the C8/C10 ratio (Line 385-394).
[ I understand that “the "C8 fluctuation range" data presented in Table 1 were measured using DBS”. You should show that in a footnote of the table. ]
Thank you for the suggestion. The "C8 fluctuation range" data presented in Table 1 were measured using DBS, and we have now added a footnote to Table 1 (Results: line 164).
[Discussion
The reference for the disease frequency in Japan as 1/51,000 (11. Shigematsu et al 2002) is too old. You should refer to the following article instead, which showed the frequency of 1/129,000 based on NBS results of 3.36 million newborns.
Shibata N, Hasegawa Y, et al. Diversity in the incidence and spectrum of organic acidemias, fatty acid oxidation disorders, and amino acid disorders in Asian countries: Selective screening vs. expanded newborn screening. Mol Genet Metab Rep 2018;16:5-10.]
This is a very valuable point. We have cited the recommended article instead (References: line 422-424 ) and revised the Discussion section accordingly (Discussion: line 219).
